# Post-Marketing Use of Teduglutide in a Large Cohort of Adults with Short Bowel Syndrome-Associated Chronic Intestinal Failure: Evolution and Outcomes

**DOI:** 10.3390/nu15112448

**Published:** 2023-05-24

**Authors:** Brune de Dreuille, Alexandre Nuzzo, Julie Bataille, Charlotte Mailhat, Lore Billiauws, Maude Le Gall, Francisca Joly

**Affiliations:** 1Department of Gastroenterology and Nutrition Support, Hôpital Beaujon (AP-HP), 92110 Clichy, France; brune.de-dreuille@inserm.fr (B.d.D.);; 2Inserm UMR 1149, Centre de Recherche sur l’Inflammation, Université Paris Cité, 75018 Paris, France; 3Inserm UMR 1148, Laboratoire de Recherche Vasculaire Translationnelle, Université Paris Cité, 75018 Paris, France; 4Pharmacy Department, Hôpital Beaujon (AP-HP), 92110 Clichy, France; 5GETAID (Groupe d’Étude Thérapeutique des Affections Inflammatoires du Tube Digestif), 75009 Paris, France

**Keywords:** intestinal failure, parenteral nutrition, short bowel syndrome, teduglutide

## Abstract

Teduglutide, a GLP-2 analogue, has been available in France since 2015 to treat short-bowel-syndrome (SBS)-associated chronic intestinal failure (CIF) but it remains very expensive. No real-life data on the number of potential candidates are available. The aim of this real-life study was to assess teduglutide initiation and outcomes in SBS-CIF patients. All SBS-CIF patients cared for in an expert home parenteral support (PS) center between 2015 and 2020 were retrospectively included. Patients were divided into two subpopulations: prevalent patients, already cared for in the center before 2015, and incident patients, whose follow-up started between 2015 and 2020. A total of 331 SBS-CIF patients were included in the study (156 prevalent and 175 incident patients). Teduglutide was initiated in 56 patients (16.9% of the cohort); in 27.9% of prevalent patients and in 8.0% of incident patients, with a mean annual rate of 4.3% and 2.5%, respectively. Teduglutide allowed a reduction in the PS volume by 60% (IQR: 40–100), with a significantly higher reduction in incident versus prevalent patients (*p* = 0.02). The two- and five-year treatment retention rates were 82% and 64%. Among untreated patients, 50 (18.2%) were considered ineligible for teduglutide for non-medical reasons. More than 25% of prevalent SBS patients were treated with teduglutide compared to 8% of incident patients. The treatment retention rate was >80% at 2 years, which could be explained by a careful selection of patients. Furthermore, this real-life study confirmed the long-term efficacy of teduglutide and showed a better response to teduglutide in incident patients, suggesting a benefit in early treatment.

## 1. Introduction

Short bowel syndrome (SBS) is a rare complex malabsorption disorder resulting from extensive intestinal resection. Indeed, an insufficient remaining small bowel length does not allow patients to meet their caloric and/or fluid requirements, leading to SBS-associated chronic intestinal failure (SBS-CIF) [1]. Home parenteral support (PS) is the reference treatment for SBS-CIF [2]. CIF may be associated with life-threatening complications (catheter-related complications, metabolic disorders, etc.) and lead to a significant impairment in quality of life [3]. SBS-CIF management requires a multidisciplinary team in an expert center, and aims at decreasing patients’ PS dependence.

In recent years, SBS management has taken a new turn with the advent of GLP-2 analogues that are now considered the first-line treatment for SBS patients in the absence of contraindications. In particular, teduglutide has been marketed in France since 2015. The ESPEN guidelines recommend the use of teduglutide in carefully selected SBS patients who are candidates for growth factor treatment [4].

GLP-2 is an endogenous hormone secreted by the enteroendocrine L cells which has effects on intestinal trophicity, barrier function, mesenteric blood flow and immune function [5,6]. As teduglutide is resistant to degradation, it increases the intestinal absorption, thus allowing a decrease in PS dependence and sometimes leading to a complete weaning from PS [7]. Several real-life studies have shown its efficacy in different countries [8,9,10,11] and have been aimed at identifying teduglutide response predictors. Treatment response is often related to SBS anatomy, the remaining bowel length and the initial PS dependence [12,13,14]. There is therefore a fair amount of data on teduglutide efficacy, although it has often been studied in the short-to-medium term.

However, little is known about the actual use of teduglutide, since its marketing authorization, in terms of patients’ selection and center-specific practices. Pironi et al. [15] have investigated the eligibility for teduglutide in a large cohort of SBS patients and divided patients into three groups: non-candidates, potential candidates and straight candidates, corresponding to patients with the lowest PS requirements and most likely to respond to treatment. The authors have highlighted the importance of describing the population of SBS patients who are candidates to GLP-2 analogues to help centers to homogeneously select patients and to compare clinical practices around the world. Furthermore, Bond et al. [16] have also emphasized the need to precisely describe the cohorts of SBS-CIF patients in the most important expert centers, and the characteristics of patients treated with teduglutide or not, in order to identify potential behaviors that could limit their eligibility.

Furthermore, this treatment is very expensive, which could explain why its indications and prescription are limited. However, until 2015, no treatment other than PS was available for SBS. Since new GLP-2 analogues will soon be available for SBS treatment, the cost of which is not yet known, and given the limited data on the target population, it seems essential to assess, in a real-life setting, which patients would be eligible for treatment. The aim of this study was thus to describe the characteristics of patients treated with teduglutide since its marketing authorization in a large cohort of SBS-CIF patients cared for in an expert center between 2015 and 2020, as well as their long-term outcomes and treatment retention rate.

## 2. Materials and Methods

### 2.1. Study Design and Patients

This was a monocentric, observational, retrospective study conducted in the Department of Gastroenterology of Beaujon University Hospital (Clichy, France), an expert center caring for all patients with CIF in Ile de France.

All SBS-CIF adult patients followed in the center between 2015 and 2020 were included, given that 2015 is the year teduglutide was granted its marketing authorization. Inclusion criteria were patients with a diagnosis of SBS (remnant small bowel length < 250 cm) and PS dependence (parenteral nutrition and/or fluid therapy) at the time of inclusion. The inclusion date was 1 January 2015 for patients diagnosed with SBS before 2015, or the date of diagnosis for patients diagnosed with SBS between 2015 and 2020. The baseline date was defined as this inclusion date.

Patients were followed until 31 December 2020, their death, or the date the latest information was obtained in patients lost to follow-up, whichever came first. Patients were divided into two subpopulations: the prevalent population, composed of patients who were already cared for in the center before 2015, and the incident population, composed of patients diagnosed with SBS after 2015. All SBS-CIF patients who initiated treatment with teduglutide between 2015 and 2020 were included in the study.

#### 2.1.1. Primary Outcome

The evolution of teduglutide prescription was investigated in a cohort of SBS-CIF patients after its marketing authorization (2015).

The incidence of treated patients over time was assessed based on the new prescriptions on an annual basis between 2015 and 2020 to calculate a mean annual rate of teduglutide initiation in the prevalent and incident populations.

#### 2.1.2. Secondary Outcomes

Treatment efficacy, as well as the long-term exposure and characteristics of treated patients, were assessed.

The efficacy of teduglutide was assessed based on the reduction in PS (% and volume (mL/week)) achieved during treatment compared to the time of teduglutide initiation. The percentage of weaned patients was calculated in both populations.

Teduglutide retention rate was calculated at 6 months and then on an annual basis. The reasons for teduglutide discontinuation were identified from the medical charts.

Clinical characteristics of teduglutide-treated patients were compared to those of untreated patients. The reasons for treatment ineligibility were identified from the medical charts with regard to contraindications, or reported by physicians (JF, LB) with regard to declarative reasons.

### 2.2. Data Collection

Demographics and clinical data were collected from the medical charts, including the age, gender, initial disease that led to intestinal resection (mesenteric ischemia, Crohn’s disease, chronic intestinal pseudo-obstruction (CIPO), radiation enteritis, cancer, surgery complications), bowel anatomy features (remnant bowel length and colon in continuity, type of stoma, reverse intestinal interposition surgery), comorbidities (hypertension, renal failure, heart disease, dyslipidemia, diabetes), and information on PS dependence (volume, number of infusions and energy). Data were collected at baseline and throughout the follow-up period. For patients treated with teduglutide, specific information on treatment was collected throughout the treatment period, including efficacy (change in PS), tolerance (identification of adverse effects: abdominal pain, nausea, injection site reactions, stoma-related events, diarrhea, tiredness), and treatment discontinuation (occurrence of a contraindication, poor tolerance, no significant benefit).

### 2.3. Statistical Analysis

Continuous data are presented as a median and interquartile range (IQR), and a Wilcoxon–Mann–Whitney test was used for comparisons, given that the variables were not normally distributed. Categorical data are presented as a number of patients (percentage of patients) and were compared using a Pearson’s Chi2 test or a Fisher’s exact test, as appropriate. A *p* value < 0.05 was considered significant. The teduglutide retention rate was calculated using a Kaplan–Meier survival estimate. A two-way ANOVA was used to compare the evolution of PS in the prevalent and incident populations during treatment with teduglutide. All analyses were performed using R studio software (RStudio, PBC, Boston, MA URL), version 4.2.1.

### 2.4. Ethical Approval

The study was approved by our institutional review board (number 00006477). A dedicated case report form (CRF) was created (Doqboard). All patients were entered in this software using an anonymized identification.

## 3. Results

### 3.1. Characteristics of SBS-CIF Patients

A total of 331 patients were included in the study. The median age at the time of inclusion was 54.6 years (IQR: 42–69 years), and 186 (56%) patients were female. The colon was in continuity in 57% of patients (45% with a jejunocolic anastomosis and 12% with a jejunoileal anastomosis and an intact colon). The characteristics, and a comparison of the prevalent and incident populations, are presented in Table 1. The main difference between both populations was the bowel anatomy; prevalent patients had a shorter remnant small bowel (*p* < 0.001) and had undergone fewer ileostomies than incident patients (2% versus 18%, *p* < 0.001). The causes of SBS also differed between both populations (more cases of CIPO and fewer cases of cancer in the prevalent population), as well as the prognosis of patients (incident patients had a better prognosis).

### 3.2. Evolution of Teduglutide-Treated Patients over Time

Among the SBS patients included in this study, 56 (16.9%) initiated treatment with teduglutide: 27.9% and 8.0% of prevalent and incident patients, respectively. Figure 1 shows the evolution of the cumulative number of patients treated with teduglutide each year between 2015 and 2020 in the prevalent and incident populations. The number of treated prevalent patients increased until 2017, and then tended to reach a plateau. The incident patients initiated treatment later, but the number of treated patients kept increasing in this subpopulation. The mean annual rate of teduglutide initiation each year between 2015 and 2020 was 3.4% in the whole cohort: 4.3% in the prevalent population and 2.5% in the incident population. Table 2 shows the percentage of patients who initiated teduglutide each year in both subpopulations. This percentage was especially high during the first three years following teduglutide marketing (up to 9% of prevalent patients initiated treatment in 2016), and was much lower between 2018 and 2020. Table 2 also reports the median duration of PS before initiating treatment each year in both subpopulations.

### 3.3. Long-Term Efficacy of Teduglutide

In our cohort of SBS patients, teduglutide allowed a decrease in PS volume by 60% (IQR: 40–100%) compared to baseline, and allowed weaning for 20 patients (36%) from PS, after a median treatment duration of 2.7 years (IQR: 1.7–4.4 years). Figure 2 shows the median evolution in PS volume (mL/week) during treatment in the prevalent and incident populations. Although all patients received a similar volume of PS at the time of teduglutide initiation, incident patients received a lower volume of PS (mL/week) during treatment than prevalent patients (*p* = 0.02), showing a stronger decrease in PS volume compared to baseline. Thus, incident patients appeared to better respond to teduglutide. Furthermore, 7 incident patients (50%) were totally weaned from PS during treatment, compared to 13 prevalent patients (31%).

### 3.4. Teduglutide Retention and Discontinuation Rates

The treatment retention rate over time in this cohort is shown in Figure 3. The retention rate was 89% at 6 months, 84% at 1 year, 82% at 2 years, and 64% at 5 years. Overall, 19 patients (34%) discontinued treatment, after a median duration of 27.0 months (IQR: 4.9–29.9 months).

The reasons for teduglutide discontinuation are presented in Table 3. The main causes were the occurrence of contraindications (*n* = 6, 31.6%), adverse events (*n* = 5, 26.3%), and a lack of efficacy (*n* = 4, 21.1%). There were no clinical differences between patients who discontinued treatment and those who did not.

### 3.5. Assessment of Teduglutide Prescription Criteria

The characteristics of treated patients and the reasons for treatment ineligibility were assessed.

The baseline characteristics of patients who initiated treatment with teduglutide and those who did not are summarized in Table 4. Treated patients were younger than untreated patients (*p* = 0.007), and initiated PS at a younger age (*p* < 0.001). Moreover, they had a shorter remnant small bowel (*p* = 0.01), and a cancer was not the cause of their SBS (unsurprisingly, cancer being one of the contraindications to treatment). However, they showed similar characteristics to untreated patients in terms of intestinal anatomy (presence of the colon, jejunostomy) and SBS causes apart from cancer.

For the 275 patients of the cohort who were not treated with teduglutide during the study period, the causes of ineligibility for treatment were identified. The main reasons were a history of cancer in the last 5 years (*n* = 54, 19.6%), spontaneous weaning from PS (*n* = 40, 14.5%), and the presence of comorbidities (*n* = 36, 13.1%). However, 18.2% of patients were considered ineligible for treatment for non-medical reasons, including the patient’s refusal (*n* = 17, 6.2%), a doubt about the patient’s compliance (*n* = 10, 3.6%), the presence of a cognitive disorder (*n* = 8, 2.9%) or other reasons, including language barrier, complicated follow-up or inadequate environment.

## 4. Discussion

This was the first real-life study to assess the evolution of the number of SBS patients initiating treatment with teduglutide and their characteristics since 2015. Two subpopulations of SBS-CIF patients could be distinguished: SBS patients who were already PS-dependent, possibly for a long time, when teduglutide was granted its marketing authorization (the prevalent population), and patients who developed SBS-CIF and became PS-dependent in a context where teduglutide was already available and used in clinical practice (the incident population). In our cohort, the prevalent population had a poorer prognosis, this result being consistent with the fact that the prevalent population included patients treated with long-term PS, who were likely to have more comorbidities than the incident population in which some patients were rapidly weaned from PS.

The number of patients treated with teduglutide was higher in the prevalent versus the incident population, consistent with the fact that patients with long-term stable PS are the target population for this treatment, and they were thus the first patients to receive it when it was marketed. Indeed, from 2015, the number of prevalent patients under teduglutide increased strongly until 2017, then tended to reach a plateau, whereas the number of treated incident patients was initially low and kept increasing until 2020. Thus, incident patients progressively replaced prevalent patients as the target population. However, some prevalent patients still initiated treatment in the last years of the follow-up, and this could be due to the fact that some patients need more time and hindsight regarding treatment before accepting it. This finding supports the importance of informing patients of the treatment outcomes in a real-life setting and encouraging interactions between patients to share their experiences.

The assessment of the percentage of patients initiating treatment each year in both populations showed that teduglutide was more prescribed in the first three years after its marketing, compared to the subsequent period until today. This result is noteworthy because it means that the use of teduglutide was very important at a time when little experience was available. It is essential to promote structured protocols and training programs to avoid complications and optimize long-term teduglutide retention.

The proportion of patients treated with teduglutide in our cohort could be extrapolated to estimate the number of patients who will benefit from teduglutide in a center starting to prescribe it. For instance, treatment with teduglutide may now be prescribed in Denmark. In an international multicenter survey, conducted in 2015, of 2919 adult patients requiring home PS [17], 64.3% of them presented with SBS (1877 patients). Among the patients included, 233 were registered in Denmark. Considering our rate of 27.9% of treated patients in the prevalent population, we could assume that about 65 of these patients will be candidates for teduglutide within five years.

In terms of efficacy, treatment with teduglutide was effective in the long term to reduce PS dependence in our SBS patients, with a significant decrease in PS volume in the first six months of treatment. This study confirmed, in the longer term, the results of five real-life studies that have reported a percentage of response of 71–88% after 6–36 months of treatment [8,9,10,13,18].

However, what this study added was a difference in PS reduction during treatment observed between the prevalent and incident populations, incident patients showing a better response. As incident patients were treated earlier than prevalent patients, this result suggests a potential benefit of early treatment with teduglutide. The poorer response observed in the prevalent population could also be explained by the characteristics of the patients and disease severity in this population compared to the incident population (in particular, a shorter remnant small bowel). However, prevalent patients were treated with PS for a median time of 75 months before baseline, and we could assume that the spontaneous intestinal adaptation phase had already occurred in these patients when they initiated teduglutide. On the contrary, in the incident population, early treatment could accelerate spontaneous adaptation and could even lead to a higher adaptation rate than that observed during the spontaneous plateau phase [19,20].

After two years, more than 80% of patients were still being treated. This high rate of teduglutide retention in our cohort could be due to a careful selection of treated patients. Furthermore, our center has developed a specific training program for patients and healthcare professionals with the aim to improve adhesion to the center and treatment compliance.

In most cases, treatment discontinuation was due to contraindications that appeared during treatment. The occurrence of cancers (not associated with teduglutide) suggest that it would be of interest for general practitioners to closely monitor patients, in particular based on the age and comorbidities of the patients. Moreover, some patients decided to discontinue treatment due to a lack of efficacy. In a real-life setting, it is important to consider patient reported outcomes. Even if treatment has a real benefit in terms of PS reduction, we should consider the balance between patients’ expectations and their discomfort or adverse events. Furthermore, for the young population, specific data are needed regarding a potential pregnancy. Indeed, in our cohort, we decided to discontinue treatment in two patients with a desire of pregnancy without real evidence of any contraindication. Overall, some causes of discontinuation can be relatively subjective to the patient or the clinician, and it would be interesting to compare the criteria for teduglutide discontinuation used in different centers.

We tried to characterize the types of patients selected for treatment. The comparison of treated and untreated patients in terms of clinical characteristics showed that treated patients had a shorter remnant small bowel, without any significant difference in anatomic types. This result is consistent with the fact that, in our center, we selected for treatment patients with long-term PS dependence (the prevalent population) and incident patients in whom spontaneous adaptation would not allow a weaning from PS.

A recent study has described SBS etiologies and comorbidities in patients treated with teduglutide between 2015 and 2019 from a large database of the US healthcare system [21]. Among 72 million screened patients, only 170 were prescribed teduglutide. At the national level, we could assume that the total number of patients treated with teduglutide between 2015 and 2019 was higher in France than in the US. This could be due to the centralization of care and differences in patient management between European and US centers. In France, there are 13 home PS expert centers with dedicated resources from the Health Ministry with the aim to improve coordination, multidisciplinary approaches, and training programs.

Moreover, Loufty et al. have also observed that only 10% of their patients had no ileum and no colon, while these patients are supposed to be the best responders to teduglutide (in terms of PS volume reduction). This proportion was higher in our cohort (30% of patients had undergone jejunostomy). However, 64% of our treated patients had a colon in continuity, and could therefore show a higher level of endogenous GLP-2, which could lead to a lower response to teduglutide. Nonetheless, some of the weaned patients have a colon in continuity, which is due to the fact that they often have lower PS needs upon treatment initiation. We could therefore consider that treatment could also be of benefit to these patients. Furthermore, the timing of the response to teduglutide should also be considered, and could be different depending on the anatomical type of SBS.

We assessed the reasons for not treating. Some patients were not treated due to the presence of psycho-social and environmental factors, although these patients would possibly be eligible for treatment otherwise. These limitations show the need to better manage patients to improve their commitment to a center and to try to reduce the number of untreated patients in the absence of real contraindications to teduglutide, and when spontaneous adaptation appears insufficient over time.

A similar analysis performed by other international home PS expert centers would allow the obtaining of an overall view of the use of teduglutide since its marketing authorization, comparing practices between countries and identifying potentially modifiable behaviors relating to patients’ selection. Furthermore, it would be interesting to study how these practices will evolve with the advent of new GLP-2 analogues.

In the future, it would be helpful to identify the biomarkers of treatment efficacy and toxicity. A prospective study with teduglutide dosing over time could help us understand the pharmacokinetics of the treatment and its impact on the patient’s response. Moreover, long-term studies are needed to evaluate the effect of teduglutide on cell viability. Indeed, a few studies have shown that teduglutide may have a beneficial effect on cell survival [22,23,24], but specific long-term studies are required in order to understand precisely the effect of teduglutide on enterocyte viability.

This study has several limitations, including its retrospective design, and the fact that the reasons for teduglutide discontinuation or ineligibility were collected from the information available in medical charts. In addition, several randomized trials assessing new GLP-2 analogues have been initiated in the last years [25,26], and this could reduce the number of new patients treated with teduglutide. Finally, the comparison of the prevalent and incident populations is limited by the difference in exposure to PS. In particular, the mortality rate in the prevalent population should be considered carefully because we did not include patients who developed SBS before 2015 and died before 1 January 2015 whereas, for the incident population, we included all patients. This suggests the value of creating registries to record prior deaths. However, a recent study has reported a high survival in SBS adults, with a 5-year survival of 82% in patients with non-malignant SBS [27].

## 5. Conclusions

This was the first real-life study to describe the number and characteristics of patients treated with teduglutide, since its marketing authorization, in a large cohort of SBS-CIF patients. This study confirmed the long-term benefit of teduglutide, and suggested the benefit of early treatment. The high retention rate supports the need for the multidisciplinary management of SBS patients to improve adherence to a center. This study also emphasized the need to implement national and international collaborations to obtain a global view of practices and improve patient care.

## Figures and Tables

**Figure 1 nutrients-15-02448-f001:**
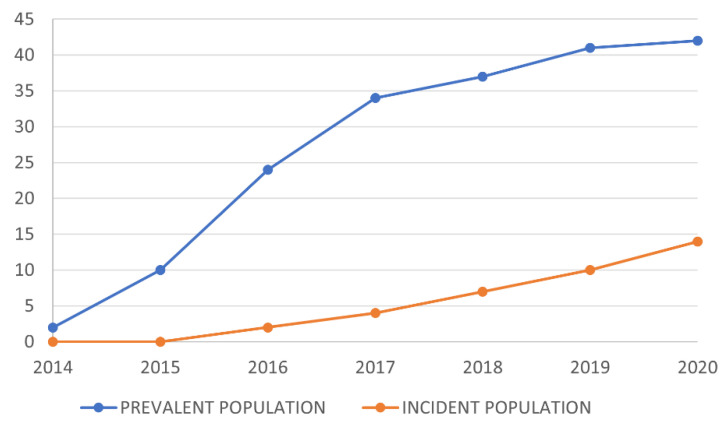
Evolution of the cumulative number of treated patients between 2015 and 2020 in the prevalent and incident populations.

**Figure 2 nutrients-15-02448-f002:**
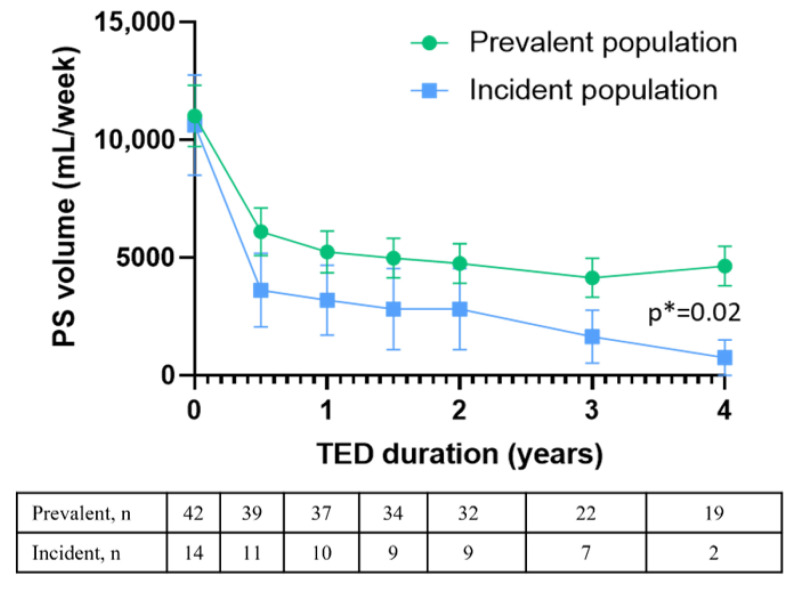
Change in PS volume during treatment with teduglutide in the prevalent and incident populations. *n* = number of patients. Dots are median values and bars represent SEM. * Two-way ANOVA.

**Figure 3 nutrients-15-02448-f003:**
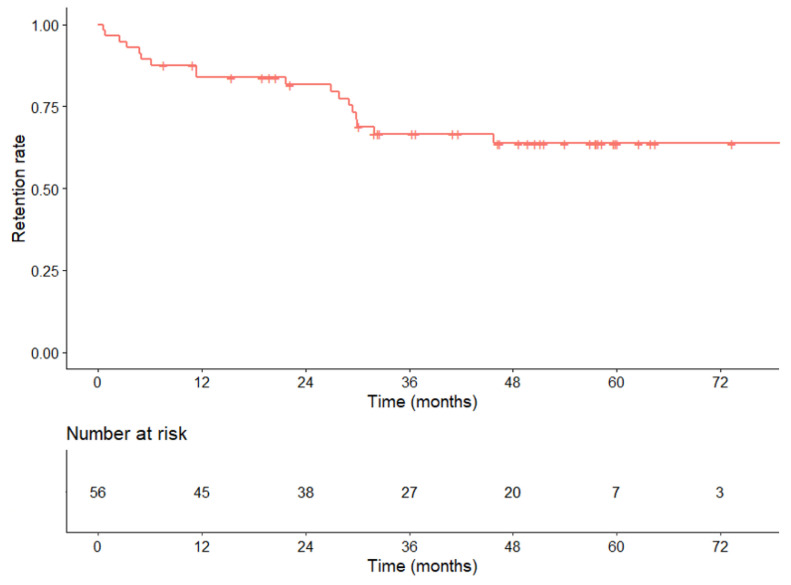
Retention rate of teduglutide treatment over time.

**Table 1 nutrients-15-02448-t001:** Comparison of the clinical characteristics of the prevalent and incident populations.

	Prevalent Population *n* = 156	Incident Population *n* = 175	*p*-Value ^(1)^
Female gender, *n* (%)	85 (54)	101 (58)	0.6314
Age at baseline (years), median (IQR)	52.0 (36.0–65.3)	61.0 (46–70)	<0.001
Age at PS initiation (years), median (IQR)	45.5 (26.0–57.5)	60.0 (44.5–69.0)	<0.001
Body weight at baseline, kg, median (IQR)	59 (52–66)	60 (50–69)	0.7497
BMI at baseline, kg/m^2^, median (IQR)	21.4 (19.4–23.5)	21.4 (18.6–24.3)	0.82
PS duration before baseline (months), median (IQR)	75 (21–102)	0	<0.001
SBS type, *n* (%)			
Type 1a	58 (37)	52 (30)	0.186
Type 1b	3 (2)	31 (18)	<0.001
Type 2	81 (52)	68 (39)	0.0229
Type 3	14 (9)	24 (14)	0.2389
Presence of ostomy	66 (42)	85 (49)	0.2408
Jejunostomy	58 (37)	74 (42)	0.6039
Colostomy	8 (5)	12 (7)	1
Remnant bowel length (cm), median (IQR)	70 (30–100)	105.4 (50–150)	<0.001
Residual colon (%), median (IQR)	50 (0–80)	50.3 (0–90)	0.1312
SBS cause, *n* (%)			
Mesenteric ischemia	49 (31)	56 (32)	1
Crohn’s disease	21 (13)	28 (16)	0.6212
CIPO	20 (13)	8 (5)	0.0078
Radiation enteritis	13 (8)	21 (12)	0.3599
Cancer	4 (3)	18 (10)	0.0067
Surgery complications	10 (6)	12 (7)	1
Other	39 (25)	32 (18)	0.2229
Comorbidities, *n* (%)			
At least one comorbidity	135 (87)	163 (93)	0.09571
Chronic renal failure	35 (22)	24 (14)	0.05414
History of cancer	35 (22)	63 (36)	0.0099
Arterial hypertension	35 (22)	59 (34)	0.0316
History of obesity	6 (4)	15 (9)	0.1249
Heart disease	0 (0)	6 (3)	0.0314
Dyslipidemia	11 (7)	23 (13)	0.1008
Outcome, *n* (%)			
Weaning from PS	28 (18)	38 (22)	0.4727
Death	36 (23)	20 (11)	0.0075
Parenteral nutrition at treatment initiation			
PS volume (mL/week), median (IQR)	9000 (6000–16,500)	10,000 (5000–17,500)	0.7151
PS calories (kcal/week), median (IQR)	6840 (4250–11,100)	6840 (3200–10,050)	0.2795
Number of days of infusion/week, median (IQR)	5 (4–7))	6 (4–7)	0.1819
Treatment with Teduglutide, *n* (%)	42 (27)	14 (8)	<0.001

Abbreviations: BMI = body mass index; CIPO = chronic intestinal pseudo-obstruction; IQR = interquartile range; PS = parenteral support; SBS = short bowel syndrome. Type 1a: jejunostomy, Type 1b: ileostomy, Type 2: jejunocolic anastomosis, Type 3: jejunoileal anastomosis. ^(1)^ Wilcoxon–Mann–Whitney test for continuous data. Fischer’s exact test (*n* ≤ 5) or Pearson’s Chi2 test (*n* > 5) for categorical data.

**Table 2 nutrients-15-02448-t002:** Percentage of patients initiating teduglutide each year between 2015 and 2020 in the prevalent and incident populations, and corresponding duration of PS before treatment initiation.

	Prevalent Population	Incident Population
Percentage of the population initiating teduglutide (%)		
2015	5.13	0
2016	8.97	5.56
2017	6.41	2.44
2018	1.92	2.7
2019	2.56	2.04
2020	0.64	2.29
Median duration of PS before teduglutide (months), median (IQR)		
2015	38.5 (21.3–124.8)	0
2016	114.5 (45.8–169.3)	11.0 (11.0–11.0)
2017	55.5 (46.5–147.8)	18.0 (15.0–21.0)
2018	76.0 (70.0–140.0)	16.5 (15.0–17.0)
2019	138.0 (134.0–165.5)	20.0 (20.0–24.5)
2020	67.0 (67.0–67.0)	11.0 (9.0–34.0)

**Table 3 nutrients-15-02448-t003:** Reasons for teduglutide discontinuation in treated patients (*n* = 19).

	*n*	%
Contraindication	6	31.6
Acute pulmonary edema	2	10.5
Diagnosis of leukemia	1	5.3
Diagnosis of melanoma	1	5.3
Diagnosis of lung adenocarcinoma	1	5.3
Severe septic shock caused by repeated bacterial translocations	1	5.3
Adverse events	5	26.3
Digestive symptoms	4	21
Injection site pain	1	5.3
No significant benefit	4	21.1
Desire of pregnancy	2	10.5
Patient’s decision	1	5.3
Poor compliance	1	5.3

*n* = number of patients.

**Table 4 nutrients-15-02448-t004:** Comparison of the clinical characteristics of treated and untreated patients.

	Treated Patients *n* = 56	Untreated Patients *n* = 275	*p*-Value ^(1)^
Age at follow-up initiation, years, median (IQR)	49.5 (35.8–61.5)	58.0 (43.5–70.0)	0.0078
Female gender, *n* (%)	26 (46)	160 (58)	0.1421
Body weight, kg, median (IQR)	62.0 (55.3–68.8)	59.0 (51.0–68.0)	0.1312
BMI, kg/m^2^, median (IQR)	21.3 (19.2–23.0)	21.4 (19.0–24.2)	0.895
PS duration before baseline (months), median (IQR)	26.7 (1.3–111.8)	0 (0–30.0)	<0.001
Age at PS initiation, median (IQR)	42 (25–55)	55 (40–67)	<0.001
Bowel anatomy features			
Type 1a, *n* (%)	17 (30)	93 (34)	0.7297
Type 1b, *n* (%)	3 (5)	31 (11)	0.232
Type 2, *n* (%)	28 (50)	121 (44)	0.4995
Type 3, *n* (%)	8 (14)	30 (11)	0.6223
Colon in continuity, *n* (%)	36 (64)	151 (55)	0.2534
Presence of ostomy, *n* (%)	24 (43)	126 (46)	0.6744
Jejunostomy, *n* (%)	20 (36)	112 (41)	0.6904
Colostomy, *n* (%)	4 (7)	16 (6)	0.5033
Remnant bowel length (cm), median (IQR)	68 (23–100)	80 (45–150)	0.0104
Residual colon, *n* (%), median (IQR)	60 (0–80)	50 (0–90)	0.637
Reverse loop, *n* (%)	4 (7)	19 (7)	1
SBS cause			
Mesenteric ischemia, *n* (%)	19 (34)	86 (31)	0.8167
Crohn’s disease, *n* (%)	12 (21)	37 (13)	0.1851
CIPO, *n* (%)	5 (9)	23 (8)	1
Radiation enteritis, *n* (%)	3 (5)	31 (11)	0.232
Cancer, *n* (%)	0	22 (8)	0.0335
Surgery complications, *n* (%)	3 (5)	19 (7)	1
Other, *n* (%)	14 (25)	57 (21)	0.6393
Comorbidities			
At least one comorbidity, *n* (%)	47 (84)	251 (91)	0.1282
Chronic renal failure, *n* (%)	15 (27)	44 (16)	0.0835
History of cancer, *n* (%)	4 (7)	94 (34)	<0.001
Arterial hypertension, *n* (%)	14 (25)	80 (29)	0.6482
History of obesity, *n* (%)	3 (5)	18 (7)	1
Heart disease, *n* (%)	1 (2)	5 (2)	1
Dyslipidemia, *n* (%)	3 (5)	31 (11)	0.232
Parenteral nutrition features			
PS volume (mL/week), median (IQR)	8800 (5000–12,500)	9885 (5909–17,500)	0.4231
PS calories (kcal/week), median (IQR)	6000 (2390–9375)	6840 (3420–10,550)	0.3666
Number of days of infusion/week, median (IQR)	5 (4–6)	6 (4–7)	0.0417
Population			
Prevalent, *n* (%)	42 (75)	114 (42)	<0.001

Abbreviations: BMI = body mass index; CIPO = chronic intestinal pseudo-obstruction; IQR = interquartile range; PS = parenteral support; SBS = short bowel syndrome. Type 1a: jejunostomy, Type 1b: ileostomy, Type 2: jejunocolic anastomosis, Type 3: jejunoileal anastomosis. ^(1)^ Wilcoxon–Mann–Whitney test for continuous data. Fischer’s exact test (*n* ≤ 5) or Pearson’s Chi2 test (*n* > 5) for categorical data.

## Data Availability

Data are available upon reasonable request to the corresponding author.

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
