# Peer review of "Post-Marketing Use of Teduglutide in a Large Cohort of Adults with Short Bowel Syndrome-Associated Chronic Intestinal Failure: Evolution and Outcomes"

_nutrients, 2023, doi:10.3390/nu15112448_

Round 1

Reviewer 1 Report

First of all, I want to congratulate the Authors on the effective and efficient SBS care program.

In the article, the Authors focused on evaluating the use of teduglutide in patients with SBS since its introduction in France.

The article was written in a clear and easy-to-read way. 

It has been divided into sections in a typical way. Their length is just right. 

Tables and schemes are appropriate. They contain data not duplicated in the manuscript. 

The adopted methodology is the main disadvantage of the presented study. It is partly adequate for research purposes (including the primary outcome - the evolution of teduglutide prescription), but on the other hand, it potentially influences some of the analysis (treatment effectiveness). The retrospective study always carries a significant risk of misinterpretation of the data. The authors noted this in the relevant paragraph of the Discussion section. On the other hand, the rare occurrence of the analyzed disease (SBS) significantly hinders the conduct of prospective studies.

The conclusions of the study are confirmed by the presented results.

The References are up-to-date and adequate to the issues discussed in the article.

The presented study does not raise ethical doubts.

Reviewer 2 Report

This study assessed the teduglutide initiation and outcomes in SBS-CIF patients. This study confirmed the long-term efficacy of teduglutide and showed a better response to teduglutide in incident patients, suggesting a benefit of early treatment

1.      All continuous data doesn't fit the normal distribution?

2.      In Figure 2, the authors only did not showed the the P-value of the interaction

3.      What about the pharmacokinetic curve of teduglutide?

4.      The authors need to add some phenotypic data.

5.      Is the teduglutide toxic for the enterocytes? The effect of teduglutide on cell viability needs to be detected.

This study assessed the teduglutide initiation and outcomes in SBS-CIF patients. This study confirmed the long-term efficacy of teduglutide and showed a better response to teduglutide in incident patients, suggesting a benefit of early treatment

1.      All continuous data doesn't fit the normal distribution?

2.      In Figure 2, the authors only did not showed the the P-value of the interaction

3.      What about the pharmacokinetic curve of teduglutide?

4.      The authors need to add some phenotypic data.

5.      Is the teduglutide toxic for the enterocytes? The effect of teduglutide on cell viability needs to be detected.

Round 2

Reviewer 2 Report

The authors should add the data of teduglutide on cell viability.
